# Fecal Supernatants from Patients with Crohn’s Disease Induce Inflammatory Alterations in M2 Macrophages and Fibroblasts

**DOI:** 10.3390/cells13010060

**Published:** 2023-12-27

**Authors:** Frida Gorreja, Mia Bendix, Stephen T. A. Rush, Lujain Maasfeh, Otto Savolainen, Anders Dige, Jorgen Agnholt, Lena Öhman, Maria K. Magnusson

**Affiliations:** 1Department of Microbiology and Immunology, Institute of Biomedicine, Sahlgrenska Academy, University of Gothenburg, 405 30 Gothenburg, Sweden; frida.gorreja@gu.se (F.G.); rushacademic@gmail.com (S.T.A.R.); lujain.maasfeh@gu.se (L.M.); lena.ohman@gu.se (L.Ö.); 2Medical Department, Randers Regional Hospital, 8930 Randers, Denmark; miabendi@rm.dk; 3Chalmers Mass Spectrometry Infrastructure, Department of Biology and Biological Engineering, Chalmers University of Technology, 412 96 Gothenburg, Sweden; otto.savolainen@chalmers.se; 4Department of Hepatology and Gastroenterology, Aarhus University Hospital, 8200 Aarhus, Denmark; andedige@rm.dk (A.D.); jorgen.agnholt@aarhus.rm.dk (J.A.)

**Keywords:** fecal metabolites, Crohn’s disease, M2 macrophages, fibrosis, inflammatory bowel disease

## Abstract

Intestinal macrophages and fibroblasts act as microenvironmental sentinels mediating inflammation and disease progression in Crohn’s disease (CD). We aimed to establish the effects of fecal supernatants (FSs) from patients with CD on macrophage and fibroblast phenotype and function. FS were obtained by ultracentrifugation, and the metabolites were analyzed. Monocyte-derived M2 macrophages and fibroblasts were conditioned with FS, and secreted proteins, surface proteins and gene expression were analyzed. M2 macrophage efferocytosis was evaluated. Patients with CD (*n* = 15) had a skewed fecal metabolite profile compared to healthy subjects (HS, *n* = 10). FS from CD patients (CD-FS) induced an anti-inflammatory response in M2 macrophages with higher expression of IL-10, IL1RA and CD206 as compared to healthy FS (HS-FS) while the efferocytotic capacity was unaltered. CD-FS did not affect extracellular matrix production from fibroblasts, but increased expression of the pro-inflammatory proteins IL-6 and MCP-1. Conditioned media from M2 macrophages treated with CD-FS modulated gene expression in fibroblasts for TGFβ superfamily members and reduced IL-4 expression compared to HS-FS. We show that M2 macrophages and fibroblasts react abnormally to the fecal microenvironment of CD patients, resulting in altered protein expression related to inflammation but not fibrosis. This implies that the gut microbiota and its metabolites have an important role in the generation and/or perpetuation of inflammation in CD.

## 1. Introduction

Crohn’s disease (CD) is a chronic inflammatory bowel disease (IBD) with unknown etiology where disease progression may result in the development of severe complications such as stenosis, fistulas and abscesses [1]. Stricturing disease, due to stenosis, is an irreversible process present already at diagnosis in around 20% of the patients, with increasing incidence over time despite anti-inflammatory treatments [1]. The cause of disease progression is unknown, but an important determinant is the functional dysbiosis seen in CD, representing altered intestinal microbiota and fecal metabolites [2,3,4]. During the disease course, the fecal microbiota is highly stable [5,6] even though temporal variability has been reported [4]. Through the production of metabolites, which are indicators of the microbial activity, the microbiota influences the intestinal immune cells such as macrophages to define the local milieu [7].

Macrophages are constantly present in the intestinal mucosa and show high plasticity depending on the environment. During steady-state, tissue-resident anti-inflammatory (M2) macrophages (M2MQs) help to maintain homeostasis, while during inflammation, newly recruited pro-inflammatory (M1) macrophages are important drivers of the inflammatory response and their numbers correlate with disease severity in patients with IBD [8,9]. Different from ulcerative colitis (UC), macrophages in CD patients with active inflammation simultaneously display a pro-inflammatory M1 macrophage phenotype and an enrichment in the expression of M2MQ-related genes [10]. Under the influence of tissue-specific signals (M-CSF, interleukin (IL) 10 and chemokine C-X3-C motif ligand 1) and ligands derived from diet and microbiota, tissue-resident macrophages become hyporesponsive to microbial stimuli and functionally accountable for elimination of apoptotic cells (termed efferocytosis) contributing to the resolution of inflammation [11,12].

The key cells in intestinal fibrosis are myofibroblasts which upon microenvironmental stimuli produce excessive amounts of extracellular matrix proteins causing fibrinogenesis [13,14]. Myofibroblasts can be generated from fibroblasts which are either replenished locally or recruited as fibrocytes to inflamed and injured tissues. Fibroblasts contribute to the healing of damage caused by an inflammation, but for unclear reasons promote fibrosis in patients with CD. In addition, under inflammatory conditions intestinal fibroblasts secrete pro-inflammatory cytokines such as IL-6 [15] and produce chemoattractants for specific immune cell subtypes [16]. However, despite the link to tissue healing, it is not clear how fibroblasts contribute to the inflammatory environment during CD. Observations suggest a direct interplay between fibroblasts, the microbiota and M2MQs where macrophages may favor fibroblast survival via growth factors and cytokines [17]. For instance, TGFβ1, a cytokine abundantly produced by M2MQs, is suggested to be involved in intestinal extracellular matrix remodeling and is linked to the fibrosis process seen in patients with CD [17]. 

In this study, we hypothesized that fecal metabolite composition impact phenotype and functions of macrophages and fibroblasts, and that the effects reflect the disease state of the fecal donor. Therefore, we aimed to determine the fecal-derived metabolite composition and the effects of fecal supernatants (FSs) on macrophages and fibroblasts, using fecal samples from patients with active CD (CD-FS) and healthy subjects (HS-FS). We also aimed to compare cell polarizing effects of FS derived from patients with CD during active inflammation with or without stenosis to elucidate if fecal factors are linked to the development of strictures. 

## 2. Materials and Methods

### 2.1. Study Subjects and Ethical Statement 

Fecal samples from patients with CD (*n* = 25) and healthy subjects (*n* = 10) (Table 1) were collected during a single-center intervention study assessing the effects of Vitamin D for patients with CD with active disease and healthy subjects [18,19]. This study was conducted at Aarhus University Hospital (Denmark) from July 2014 to October 2017 with ethical permissions from the Danish Medicine Agency (EudraCT no. 2013-000971-34), the Central Denmark Regional Committee for Health Research Ethics (no. 1-10-72-141-13) and the Danish Data Protection Agency (no. 1-16-02-296-13) [18]. For the present study, only fecal samples at baseline, prior to treatment exposure, were used. The original CD cohort consisted of 39 patients with active inflammation where 10 patients also presented with stenosis. Here, we used fecal samples from the 10 patients with stenosis and randomly selected 15 patients from the inflamed group without stenosis. For demographics and clinical characteristics, see Table 1. All fecal samples were transported between the collection center (Aarhus, Denmark) and the study center (Gothenburg, Sweden) on dry ice, stored at −80 °C degrees and thawed only once to obtain the fecal supernatants (FS).

### 2.2. Preparation of Fecal Supernatants

Fecal supernatants were obtained as previously reported [20]. Briefly, fecal samples were thawed in room temperature for 30 min and then kept on ice during the extraction process. Feces were thoroughly mixed with two weight volumes of PBS and centrifuged at 3000× *g* for 10 min at 4 °C. The supernatants were ultracentrifuged at 35,000× *g* for 2 h at 4 °C and the liquid phase, defined here as fecal supernatants (FSs), were collected. The FSs were aliquoted and stored at −80 °C.

### 2.3. Metabolomic Analysis of Fecal Supernatants 

The metabolomic investigation of fecal supernatants was conducted at Chalmers Mass Spectrometry infrastructure (Chalmers University of Technology, Gothenburg, Sweden) using gas chromatography coupled to a tandem mass spectrometer (GC-MS/MS). In short, metabolites from fecal supernatants were extracted using a mixture of water/methanol containing 10 stable isotope-labeled internal standards [21]. After drying and derivatization with oxymation and silylation, the derivatized extracts were injected into a GC-MS/MS system (Shimadzu Europa GmbH, Duisburg, Germany) and GC-MS scan data (50–750 *m*/*z*) were analyzed for targeted peak detection. Peaks detected were identified with Matlab script and data were normalized by internal standard peak intensities [22].

### 2.4. Peripheral Blood CD14^+^ Monocyte Isolation

Human blood from healthy donors in the form of buffy coats were obtained fresh the same day for isolation of CD14^+^ monocytes (Clinical Immunology and Transfusion Medicine, Sahlgrenska University Hospital, Gothenburg, Sweden; permits K 15/18 and 4/21). Peripheral Blood Mononuclear Cells were isolated using Ficoll-paque plus (GE Healthcare, Uppsala, Sweden). CD14^+^ monocytes were isolated from peripheral blood mononuclear cells using CD14 MicroBeads human isolation kit and LS columns (all from Miltenyi Biotech, Bergisch Gladbach, Germany). All protocols were performed following the manufacturers’ instructions.

### 2.5. M2MQ Polarization and FS Conditioning

Purified CD14^+^ monocytes were cultured at 2 × 10^5^ cells/well in Nuclon Delta Surface 96-well plates (Thermofisher, Roskilde, Denmark) in 200 μL serum-free base media containing 1× M-CSF (all from CellXVivo Human M2MQs Differentiation Kit, R&D Systems, Minneapolis, MN, USA) and 50 μg/mL gentamicin (Gibco, Paisley, UK). Cells were incubated at 37 °C with 5% CO_2_ for 3 days. Following the protocol, at day 3, half of the culture media were replaced with fresh media containing 1× M-CSF but with the addition of FS from HC, iCD or sCD at the final dilution 1:1000. On day 6, the media was replaced, and cells were stimulated with LPS-EK 100 ng/mL (Invivogen, Toulouse, France) in Complete Iscove’s media (Iscove’s Modifies Dulbecco’s Medium (Sigma-Aldrich, St. Louis, MO, USA), 10% fetal bovine serum (GE Healthcare Bio Sciences, Pasching, Austria), 200 mmol/L L-glutamine (Gibco) and 50 μg/mL gentamicin) and cultured for 1 day. Differentiation into a CD14^+^CD163^+^CD80^−^ M2MQ phenotype was confirmed by flow cytometry (see Appendix A), according to the manufacturer’s instructions for the M2MQ differentiation kit. Addition of FS to the cells did not alter the M2MQ phenotype. 

Supernatants were collected and frozen and cells were dissociated by washing with Hank’s Balanced Salt Solution and detaching with Enzyme-free Cell Dissociation Buffer (both Gibco) by incubating 10 min at 37 °C, 5% CO_2_. Technical replicates for both supernatants and cells were pooled and used for analyses as described below.

### 2.6. Peripheral Blood CD66b^+^ Granulocyte Isolation

For isolation of granulocytes, blood was obtained from healthy donors and processed within 10 min using a EasySep HLA Chimerism Whole Blood CD66b Positive Selection Kit using a Big Easy Separator (both from Stem Cell Technologies, Vancouver, BC, Canada) and Red Blood Cell Lysis Buffer (Roche, Mannheim, Germany). Protocols were run following the manufacturers’ instructions using PBS containing 2 mM EDTA (Sigma-Aldrich) and 0.5% FBS fetal bovine serum as isolation buffer and 5 min incubation at room temperature with Red Blood Cell Lysis Buffer at the end of the isolation.

### 2.7. Efferocytosis Assay

The efferocytosis assay was performed by using isolated CD66b^+^ granulocytes and polarized M2MQs conditioned with FS as described above. On day 0, purified CD66b^+^ granulocytes were stained with CellTrace Violet Cell Proliferation Kit (Invitrogen, Carlsbad, CA, USA) following the manufacturer’s instructions. The dye was used at a concentration of 2.5 μM and the staining reaction was stopped by adding RPMI Medium 1640 (Gibco) containing 10% FBS (GE Healthcare Bio Sciences). Spontaneous apoptosis was induced by resuspending the cells at 1 × 10^6^ cells in 1.3 mL RPMI containing 1% FBS at 37 °C and 5% CO_2_ for 24 h. To eliminate nets generated over night, the cells were resuspended in new RPMI containing 1% FBS and pre-warmed Enzyme-free Cell Dissociation Buffer, ratio 2.5:1.5 and incubated for 7 min at 37 °C, filtered and centrifuged. Cells were resuspended in Complete Iscove’s media and counted. Apoptotic status was confirmed with flow cytometry (see Flow Cytometry Analyses below). Apoptotic cells in Complete Iscove’s media were added to M2MQs, using the ratio 5:1, and incubated for 1.5 h at 37 °C and 5% CO_2_ to induce efferocytosis. Excess apoptotic cells, that had not been engulfed, were washed away with PBS. Efferocytosis was analyzed by flow cytometry as described below.

### 2.8. Glutamate Assay for Evaluation of Metabolism

Supernatants of M2MQs, conditioned with FS from HC, iCD or sCD, were used to measure glutamate as a marker for cell metabolism, using Glutamate-Glo Assay kit (Promega, Madison, WI, USA) in 96-well Half Area Assay Plates (Costar, Kenneburk, ME, USA). Following the manufacturer’s instructions, the samples were diluted 1:40, and luminescence was measured using a SpectraMax i3x luminescence multimode microplate reader (Molecular Devices LLC, San Jose, CA, USA). Glutamate consumption was related to the kit standards and FS treated samples were compared to control (M2MQs cultured without FS).

### 2.9. Primary Fibroblast Cell Line Culture

The primary colonic human fibroblast cell line CCD-18Co (ATCC, LGC Standards GmbH, Wesel, Germany) was used for all fibroblast experiments and is referred throughout the manuscript as fibroblasts. Cells were cultured in Eagle’s Minimum Essential Medium (ATCC) supplemented with FBS 10% and 50 μg/mL gentamicin. Cell culture media was changed every 2–3 days and cells were split at a sub cultivation ratio of 1:2 or 1:3. Cells were used between passages 6 and 12 when they reached approximately 80% confluence.

### 2.10. Fibroblast Assays with Fecal Supernatants or M2MQ Conditioned Media

Fibroblasts were cultured in Nunc multidish 48-well plates or Nuclon Delta Surface 96-well plates (both Thermofisher) at 20,000 cells in 400 μL media/well or 10,000 cells in 100 μL media/well, respectively. After 3 days, the media was replaced with fresh media containing FS from HC, iCD or sCD at dilution 1:1000. On day 6, cell supernatants were collected and frozen at −20 °C for cytokine/protein measurements. Cells were washed with Hanks’ Balanced Salt Solution and detached with warm trypsin-EDTA (Gibco), for 5 min at 37 °C and used for flow cytometry analysis as described below. 

Conditioned media from M2MQs stimulated with FS were produced as described above and used for priming of fibroblasts. Fibroblasts were cultured in 96-well plates at 10,000 cells in 100 μL/well. On day 3, the media was removed, and cells were cultured with 50% conditioned M2MQs media (treated with FS from HC, iCD or sCD as described above) and 50% fresh fibroblast cell culture media. On day 6, supernatants were collected and frozen at −20 °C for protein measurements. Cells were lysed with RNA Isolation RA1 Lysis Buffer (Macherey-Nagel, Dϋren, Germany) and frozen at −80 °C until RNA extraction.

### 2.11. Fibrosis Gene Expression Array

RNA from fibroblasts treated with conditioned M2MQ media was extracted using Nucleospin RNA XS extraction kit (Macherey-Nagel). Extraction was performed following the manufacturer’s instructions. From RNA, cDNA was prepared using RT2 First Strand Kit (Qiagen, Hilden, Germany) using a total of 300 ng RNA following the manufacturer’s instructions. Human Fibrosis Panel, PAHS-120ZE-1, containing 84 genes was used to detect gene expression in fibroblasts following the manufacturer’s instructions, using RT2 SYBR Green ROX qPCR Mastermix (all from Qiagen). Plates were run using a Real-Time PCR System QuantStudio 12K Flex (Applied Biosystems, Life Technologies, Waltham, MA, USA). All samples passed the quality checks for genomic DNA contamination, reverse transcription efficiency and PCR array reproducibility.

### 2.12. Protein Analyses

Supernatants of M2MQs were assayed for cytokines and other released proteins. An electrochemiluminescence multiplex assay (U-Plex, Biomarker Group 1 Assays, Meso Scale Discovery (MSD), Rockville, MD, USA) was used to simultaneously quantify M-CSF, IL-1RA, TARC, IL-6, MCP-1, Eotoxin-2, IL-23, MDC, IL-13 and IL-4. The protocol and data quantification using an MSD 1300 reader was run following the manufacturer’s instructions. Enzyme-linked immunosorbent assays (ELISAs) were used for quantitation of IL-10 using Human IL-10 uncoated ELISA kit and TGFβ1 using Human/Mouse TGFβ1 uncoated ELISA kit (both Invitrogen) in Maxisorp NUNC-Immuno flat bottom 96-well plates (Thermofisher). 

Fibroblast supernatants were assayed for Pro-collagen I using DuoSet ELISA Human Pro-Collagen I a1/COLIA1 kit, fibronectin using DuoSet Human Fibronectin and IL-6 using DuoSet Human IL-6 ELISA kit with the DuoSet Ancillary Reagent Kit 2 (all from R&D Systems).

### 2.13. Flow Cytometry Analyses

For surface marker staining, M2MQs were first incubated with Fixable Aqua Dead Cell Stain Kit (Invitrogen, Eugene, OR, USA) for 15 min. Cells were then washed with FACS buffer (PBS containing 3% FBS, 15 mM HEPES (Gibco) and 5 mM EDTA) and incubated with Anti-Hu Fc Receptor Binding inhibitor purified (Invitrogen) for 10 min. Next, cells were stained with a cocktail of antibodies containing anti-CD14 (APC-H7), anti-CD163 (BV711), anti-CD206 (BB515), anti-CD64 (PeCy7) (all BD Biosciences, Franklin Lakes, NJ, USA), anti-CD11c (BV421) (Invitrogen) and anti-CD36 (APC) (Molecular Probes, Fredrik, MD, USA) for 20 min. Analysis was performed by gating subsequently as follows: cells (FSC-A vs. SSC-A), single cells (FSC-A vs. FSC-H), live cells (Fixable Aqua Dead Cell Stain negative) and finally specific cell markers. Purified CD66b+ granulocytes were stained with CellTrace Violet (as described above). Apoptosis was confirmed after 24 h by staining with Annexin V (APC) and 7AAD in Annexin binding buffer (0.01 M Hepes (pH 7.4), 0.14 M NaCl, 2.5 mM CaCl_2_). AnnexinV^+^7AAD^−^ cells were considered apoptotic. 

Efferocytosis was evaluated by quantifying CellTrace Violet (% and median fluorescent intensity) of M2MQs identified by anti-CD14 (APC-H7) and anti-CD163 (BV711), staining was performed as described above. 

Fibroblasts were stained for intracellular and surface markers as follows. Cells were stained with a surface marker cocktail using anti-Fibroblast Activation Protein (FAP) (PE) (R&D Systems), anti-CD21 (Pe-Cy7) and anti-PD-L1 (BV786) (both BD Biosciences), by incubating for 20 min at 4 °C. Afterwards, cells were washed with FACS buffer, fixed for 10 min, washed and permeabilized for 10 min using Fixation and Permeabilization Buffer Kit (R&D Systems). Finally, cells were stained intracellularly with anti-α-smooth muscle actin (α-SMA) (APC) (R&D Systems) and anti-TGFβ1 (BV421) (BD Biosciences) for 20 min. 

Cells were analyzed using flow cytometers LSRII or LSR Fortessa X-20 (both from BD Biosciences) and cells were kept on ice, in the dark, during all staining procedures. Gates were set using fluorescence minus one (FMO) control.

### 2.14. Biostatistics and Data Analysis

Data analysis was performed using FlowJo 10.7.2 (Ashland, OR, USA), R 3.6.0 [23], GraphPad Prism 9 (GraphPad Software, San Diego, CA, USA) and MetaboAnalyst 5.0 [24]. The level for significance was set to <0.05. For the metabolomics data, Pathway Analysis and Statistical (one factor) analysis was performed using MetaboAnalyst 5.0 online website (https://www.metaboanalyst.ca). For all other tests between the two groups, the Mann–Whitney U test was used. The chi-square test was used to analyze the differences between categoric variables for patient demographic data. PCAs were performed using the pca3d package in R, after log10 transformation and centering of the data. 

The fibrosis array gene expression data were analyzed using R Packages; for data formatting, dplyr, magrittr, tidyr, readxl, stringr, and purrr were used; for data analyses, emmeans were used; and for visualization, ggplot2 was used [25,26]. Summary statistics of the geometric means of normalized gene expression are presented. Differential expression was assessed via linear models with log2 (normalized gene expression) as response adjusted for treatment and a growth factor score as covariate using growth factor genes AGT, CCN2, EDN1, EGF, HGF, PDGFA, PDGFB and VEGFA. The adjustment for growth factor score was carried out to account for cells being in different cell cycle stages. Normalized gene expression was computed as the ratio of gene expression divided by the geometric mean of housekeeping gene expression (ACTB, B2M, GAPDH, HPRT1, RPLP0). The growth factor score is computed as the ratio of the geometric mean of growth factor genes expression divided by the geometric mean of house-keeping gene expression. A separate model was fit for each gene and each comparison (HC vs. CD, iCD vs. sCD); *p*-values for pairwise comparisons were derived from these model-based t-tests. All authors had access to the study Matlab script and data were normalized by internal standard peak intensities [22].

## 3. Results

### 3.1. Patients with CD Have a Skewed Profile of Fecal-Derived Metabolites

To determine the metabolite composition of FS, metabolomic profiling was performed and a principal component analysis (PCA) based on 109 metabolites suggested differences in fecal-derived metabolites between healthy subjects (HS *n* = 10), patients with CD with active inflammation (iCD, *n* = 15) and patients with CD with active inflammation and stenosis (sCD, *n* = 10) (Figure 1A,B). Abundance testing exhibited a 2-fold increase for 17 metabolites in CD (iCD + sCD) as compared to HS (Figure 1C), while for iCD compared to sCD only two metabolites, N-Acetylornthine and Fucose, were increased 2-fold (Figure 1D), which also appeared when comparing CD vs. HS. Pathway analysis for the 17 metabolites that differed between CD and HS revealed significant impact in three pathways: arginine biosynthesis, alanine aspartate and glutamate metabolism and arginine and proline metabolism (Figure 1E).

### 3.2. Fecal Supernatants from Patients with CD Condition Macrophages to Express a Distinct anti-Inflammatory Protein Profile upon LPS Stimulation

To investigate the effects of FS on macrophages, primary CD14^+^ monocytes were matured into anti-inflammatory CD14^+^CD163^+^CD80^−^ M2MQs for 6 days with macrophage colony-stimulating factor (M-CSF). During days 3–6, FSs were added at a dilution of 1:1000, and on day 6, the cells were stimulated with lipopolysaccharide (LPS) (Figure 2A). A PCA based on 12 secreted M2MQ characteristic proteins (Eotaxin-2, IL-13, IL-23, IL-4, IL-6, M-CSF, MDC, TARC, MCP-1, IL-RA1, IL-10 and TGFβ1) showed some separation between the groups (Figure 2B). In detail, IL-10, TGFβ2, IL1RA and IL-6 were increased in CD-FS compared to HS-FS (Figure 2C,E–G). TGFβ1 was increased and TGFβ2 decreased in iCD-FS as compared to sCD-FS (Figure 2D,E). The other proteins, including M-CSF (Figure 2H), were similar between the groups.

Next, we evaluated the expression of key surface marker proteins (for gating see Appendix A) and expression of CD206, CD36 and CD11c was increased on M2MQs treated with FS from patients with CD as compared to HS (Figure 3A–C). When comparing iCD-FS to sCD-FS, CD206 was increased (Figure 3A) and CD36 tended to increase (Figure 3B) in sCD-FS. CD64 expression did not differ between any of the groups (Figure 3D). Taken together, FS from patients with CD induced a more pronounced anti-inflammatory response in M2MQs after LPS stimulation as compared to healthy FS. Only minor differences were detected between iCD-FS and sCD-FS.

### 3.3. Efferocytotic Function of Macrophages Is Unaltered after Exposure to Fecal Supernatants

Given the changes in secreted proteins and expression of phenotype markers upon exposure to FS, we investigated its impact on efferocytosis. Fluorescently labelled apoptotic granulocytes were added to FS-treated M2MQs and efferocytosis was analyzed by flow cytometry (Figure 4A). Apoptotic granulocytes were defined as AnnexinV^+^7AAD^−^ (Figure 4B) and efferocytosis was assessed among CD14^+^CD163^+^ M2MQs (Figure 4C). Results showed that FS, regardless of study group, did not alter the efferocytotic function of M2MQs (Figure 4D). In addition, as a measure of macrophage metabolism, glutamate was measured in culture media from M2MQs to analyze possible alterations in energy production. Glutamate levels did not differ between M2MQs stimulated with FS from either HS compared to CD or iCD compared to sCD (Figure 4E).

### 3.4. Fecal Supernatants from Patients with CD Induce Pro-Inflammatory Protein Expression in Fibroblasts

To investigate the effect of FS on fibroblasts, we cultured the colonic human fibroblast cell line CCD-18Co with FS from HS and CD, diluted 1:1000, as shown in Figure 5A. Results showed higher secretion of IL-6 and MCP-1 for fibroblasts treated with FS from patients with CD as compared to HS (Figure 5B), while no differences were detected between iCD and sCD (Figure 5C).

### 3.5. Fecal Supernatants Do Not Induce Fibrosis-Related Alterations in Fibroblasts

We then investigated if FS could induce pro-fibrotic alterations in fibroblasts (setup as shown in Figure 5A). Pro-collagen I alfa 1 (Pro-COL1A1) secretion, a precursor of the most abundant extracellular matrix component was not affected by FS treatment regardless of the group (Figure 6A). Next, a number of cell surface markers related to fibrosis were evaluated. We defined the fibroblasts as stable α-SMA^high^, and FS treatment did not alter α-SMA expression (Figure 6B,C). Treatment with HS-FS, CD-FS, iCD-FS and sCD-FS did not show differences in FAP or CD21 expression (Figure 6D,E), whereas PD-L1 expression was upregulated for CD-FS compared to HS-FS (Figure 6F).

### 3.6. Conditioned Media from M2MQs Treated with FS from Patients with CD Induce Alterations of TGFβ Superfamily-Related Genes and IL-4 in Fibroblasts

Since no fibrosis-related effects of FS were detected, we next tested if conditioned media from FS-treated M2MQs could impact fibroblast function. Thus, fibroblasts were cultured with conditioned media from M2MQs for 3 days (Figure 7A). There was an increased expression of pro-COLA1, and a decreased expression of CD21 for iCD-FS as compared to sCD-FS, but no differences between HS-FS and CD-FS (Table 2). The secretion of fibronectin and expression of fibroblast surface markers, FAP and PD-L1 and intracellular staining for TGFβ1 was similar for all groups (Table 2). Finally, we investigated gene expression of fibroblasts cultured with M2MQ conditioned media using an rtPCR array containing 84 genes related to fibrosis. Comparison between HS-FS and CD-FS resulted in 15 differentially expressed genes belonging to seven different families with the TGFβ superfamily members being most prominent (Figure 7B). There was a considerable decrease in IL-4 expression in cells treated with FS from patients with CD as compared to HS (Figure 7B). Comparison of iCD-FS with sCD-FS only showed differential expression of the TGFβ2 gene (Figure 7C).

## 4. Discussion

In this study, we show that FS from patients with active CD compared to FS from HS promote an anti-inflammatory protein profile in M2MQs and a pro-inflammatory protein profile in fibroblasts. Furthermore, FS from sCD induced more pronounced anti-inflammatory effects in M2MQ than iCD, while the pro-inflammatory changes in fibroblasts were comparable. Finally, conditioned media from M2MQs cultured with FS from patients with CD modulated gene expression of members from the TGFβ superfamily and IL-4 in intestinal fibroblasts.

Similar to a previous study, we demonstrated that patients with CD have a skewed fecal metabolomic profile [27], and pathway analysis highlighted that this impacts the arginine biosynthesis pathway, alanine aspartate and glutamate metabolism and arginine and proline metabolism. An increase in fecal metabolites from these pathways might be an indication of impaired metabolite uptake by intestinal cells, which is supported by the report of decreased serum levels of arginine and glutamine in patients with CD [28]. Indeed, the attempt to dampen experimental colitis by high oral doses of arginine instead resulted in more intestinal inflammation and increased collagen deposition [29]. Interestingly, metabolization of arginine by macrophages gives rise to both nitric oxide and urea, which are molecules reported to have detrimental roles in balancing pro- and anti-inflammatory mechanisms in IBD pathology [30]. Altered abundance of metabolites may therefore impact the ability of macrophages to fine-tune this balance. 

Anti-inflammatory M2MQs challenged with LPS produce anti-inflammatory proteins that help dampen inflammation, promote tissue repair and wound healing [31]. To mimic the conditions in the intestinal microenvironment, M2MQs were first allowed to start polarizing with M-CSF and only thereafter FSs were added to contribute to the final polarization. M2MQs treated with FS from patients with CD produced higher levels of several key regulators of inflammation, including anti-inflammatory secreted cytokines and surface bound receptors expressed by tissue-resident macrophages. This suggests that M2MQs treated with FS from patients with CD derive a regulatory profile to counteract the pro-inflammatory processes in the CD intestinal microenvironment. Parallel to this, we have previously shown that M1 macrophages treated with FS from patients with ulcerative colitis increased IL-10 and pro-inflammatory cytokine secretion [20]. In addition, expression of IL-10 is higher in intestinal areas with intense inflammation as compared to regions with less inflammation [32], and it is known that macrophages are a predominant source of IL-10. Tissue-resident macrophages are required for intestinal mucosal wound healing [33] and, upon maturation, e.g., CD206 expressing colonic macrophages, seem to acquire higher expression of IL-10 [34]. CD-FS also induced a modest increase in CD11c on M2MQs, although the expression was still low. CD11c is a marker which is highly expressed by pro-inflammatory macrophages, absent from tissue-resident macrophages, and CD11c^dim^ cells are described as a transition phenotype between the two [35]. Possible implications FS effects on CD11c on M2MQs still warrant further investigation. 

Regarding the function of M2MQs, neither efferocytotic function, as an indirect measure of resolution of inflammation, nor the glutamine consumption, as a measurement of alteration in metabolism which is seen during reprogramming of macrophages [36], were affected by FS treatment. Efferocytosis in macrophages is inhibited by LPS, an effect showed to be counteracted by IL-10 [37]. Furthermore, CD36, a receptor for uptake of apoptotic cells, promotes efferocytosis [38]. Despite an altered polarization of the M2MQs, including increased secretion of IL-10 and expression of CD36 and CD206, driven by the FS from patients with CD, net efferocytosis was unaltered. Secreted proteins were not analyzed in our setting since no major effects could be expected due to the short incubation time (1.5 h). Thus, how the downstream effects of efferocytosis, e.g., by the production of pro-resolving factors or cytokines supporting resolution, differ between CD and HS in vivo in relation to the microbiota remains to be determined. 

Here, we show that FS from patients with CD compared to healthy FS induced a more pronounced pro-inflammatory response in fibroblasts, suggesting that the fecal microenvironment in patients with CD shape fibroblasts to contribute to the ongoing inflammation. Colonic fibroblasts from patients with CD have previously been shown to express MCP-1 and IL-8 [39] and overexpress IL-6 in response to LPS [40]. Potentially, an increased secretion of IL-6 from fibroblasts, may support the T helper (Th) 17 cell differentiation reported in CD [41]. Concerning tissue structure- and fibrosis-related proteins, we did not detect any FS-related alterations for α-SMA, FAP, CD21 or collagen, all known to be related to fibroblasts and fibrosis [42,43,44]. However, stimulation of fibroblasts with CD-FS increased PD-L1 expression, a marker shown to be associated with pulmonary fibrosis in a humanized mouse model [45]. Still, no differences were detected when comparing iCD-FS and sCD-FS which could link our findings to the presence of a microbial fibrotic microenvironment. 

Since an increased number of tissue-resident macrophages are found in and around fibrotic lesions [41], we cultured fibroblasts with conditioned media from FS-treated M2MQs to mimic their interaction in the fibrotic microenvironment. An over-representation of altered expression of genes from the TGFβ superfamily was detected between CD-FS and HS-FS. The functions of the proteins from this superfamily are diverse and related to many different cellular processes, thereby making the downstream effects difficult to predict. Interestingly, there was a strong decrease in IL-4 expression for CD-FS vs. HS-FS. Since IL-4 is important for a Th2 response, a reduction in IL-4 together with the abovementioned induction of IL-6 suggests a shift from a Th2 response towards a Th17 response in CD which could be driven by the metabolites. This needs to be confirmed in T cell differentiation experiments, but several studies report that the microbiota from patients with CD promotes Th1/Th17 responses over Th2 [46,47]. 

It may seem contradictory that opposite effects are detected by FS treatment on M2MQs and fibroblasts. Potentially, the M2MQ polarization in vitro allows for only limited plasticity of cells as compared to the in vivo situation, and FS may enhance already activated pathways instead of shifting phenotype and function. The fibroblast cell line in our culture models was α-SMA^high^, an indication of tissue myofibroblast phenotype, but may still react differently when taken out of context, i.e., the intestinal tissue. To add to this, we have shown that FS from patients with ulcerative colitis augment pro-inflammatory effects of M1MQs [20] and induce skewed gene expression profiles in epithelial Caco2 cells and colonoids [48]. Thus, FS has different effects on diverse type of cells but the net effects on the intestinal mucosa in vivo are currently unknown. Importantly, the regulation of phenotype and function of cells is also donor/patient specific. 

We recognize several limitations of this study. The number of study subjects were relatively low, but the patient groups were well defined and well characterized. Still, we were not able to study stenosis alone since active CD patients with stenosis also had inflammation. Monocyte-derived M2MQs will only partially reflect tissue-resident macrophages but the latter are low in abundance in the intestine and would be too few for these types of studies [49]. Similarly, the use of an in vitro intestinal fibroblast cell line is a limitation. However, designing a model, to reflect the fibrotic (stenosis) FS microenvironment, with fibroblasts and macrophages is difficult considering the limited understanding of the complex microenvironmental signals. Further, fibrosis is a long-term process which is difficult to capture in short-term models, and we acknowledge that our cell models are simplifications of the in vivo situation but they still allow us to determine distinct effects of FS from patients with CD compared to HS. Finally, even though we show that CD patients have increased amounts of certain fecal metabolites, we cannot exclude other factors contained in FS, such as proteins, lipids or different PAMPs, as being responsible for the effects observed on our in vitro culture models. 

## 5. Conclusions

In conclusion, our results suggest that the fecal microenvironment provides fine-tuning signals to macrophages and fibroblasts to keep intestinal homeostasis and maintain an immunological balance that is lost in CD. The over-active anti-inflammatory action of macrophages and induced pro-inflammatory effect from fibroblasts caused by microenvironmental factors in CD may be involved in perpetuating the disease. The results also suggest that fibroblast contribution to the ongoing inflammation in CD goes beyond generation of fibrotic tissue. Together this implies that treatments targeting the microbiota, could be promising to counteract the chronicity of the disease. 

## Figures and Tables

**Figure 1 cells-13-00060-f001:**
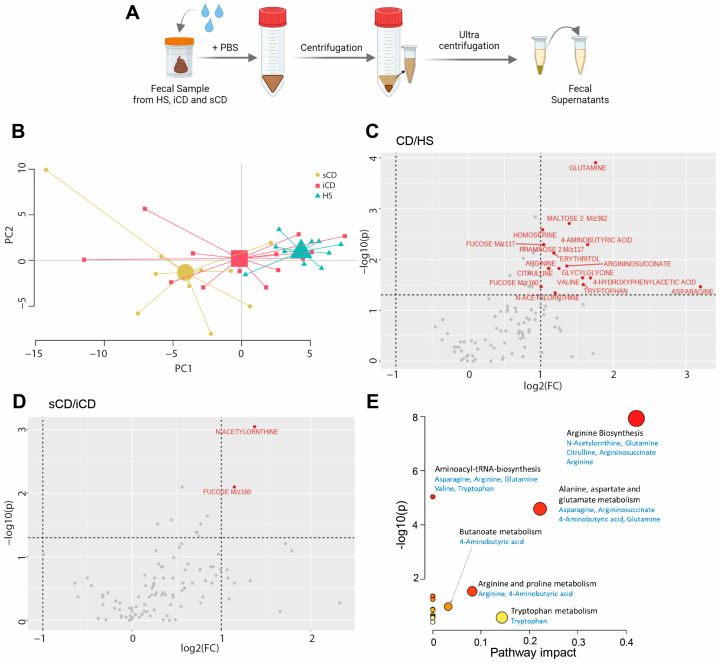
The fecal metabolome in patients with CD and HS. Metabolomic analysis was performed in fecal supernatants using GC-MS/MS for CD patients with inflammation (iCD, *n* = 15), CD patients with inflammation and stenosis (sCD, *n* = 10) and healthy subjects (HS, *n* = 10). (**A**) Schematic overview of obtaining fecal supernatants from fecal samples. (**B**) Principal component analysis showing fecal metabolite abundance levels based on 109 metabolites. Volcano plots showing log2 fold change (FC) and *p*-value of metabolite peak intensities comparing (**C**) CD with HS and (**D**) sCD with iCD. Significantly altered metabolites in panel (**C**,**D**) with more than a two-fold change are shown in red (*p* < 0.05), all others are shown in gray. Significance was assessed by One-way analysis of variance (ANOVA) via MetaboAnalyst. (**E**) MetaboAnalyst pathway analysis of significantly different metabolites between CD and HS. The bigger the circle, the higher the pathway impact. The redder the circle, the lower the *p*-value.

**Figure 2 cells-13-00060-f002:**
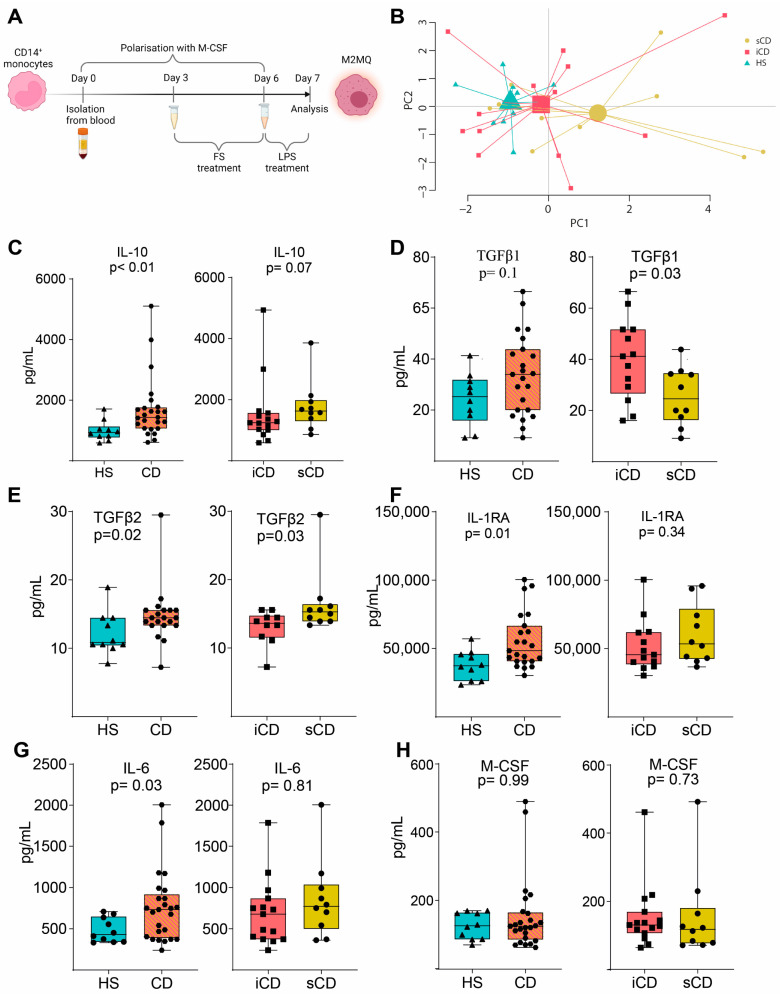
LPS-stimulated M2MQs treated with FS from patients with CD and HS. (**A**) CD14^+^ monocytes from healthy blood donors were matured into CD14^+^CD163^+^CD80^−^ macrophages with M-CSF for 6 days, with or without FS, diluted 1:1000, during days 3–6. On day 6, cells were stimulated with 100 ng/mL LPS overnight. FS from CD patients with inflammation (iCD, *n* = 15), CD patients with inflammation and stenosis (sCD, *n* = 10) and healthy subjects (HS, *n* = 10) were used. Protein secretion was measured in the supernatants using MSD and ELISA. (**B**) Principal component analysis of proteins (Eotaxin-2, IL-13, IL-23, IL-4, IL-6, M-CSF, MDC, TARC, MCP-1, IL-RA1 IL-10, TGFβ1) from stimulated macrophages with FS from iCD, sCD and HS. LPS-induced IL-10 (**C**), TGFβ1 (**D**), TGFβ2 (**E**), IL-RA (**F**), IL-6 (**G**) and M-CSF (**H**) secretion in supernatants from macrophages treated with FS from patients with iCD, sCD and HS. Significance was calculated by the Mann–Whitney U test. Graphs where a *p*-value is not shown are not significant. For each biological replicate, technical duplicates were pooled before analysis.

**Figure 3 cells-13-00060-f003:**
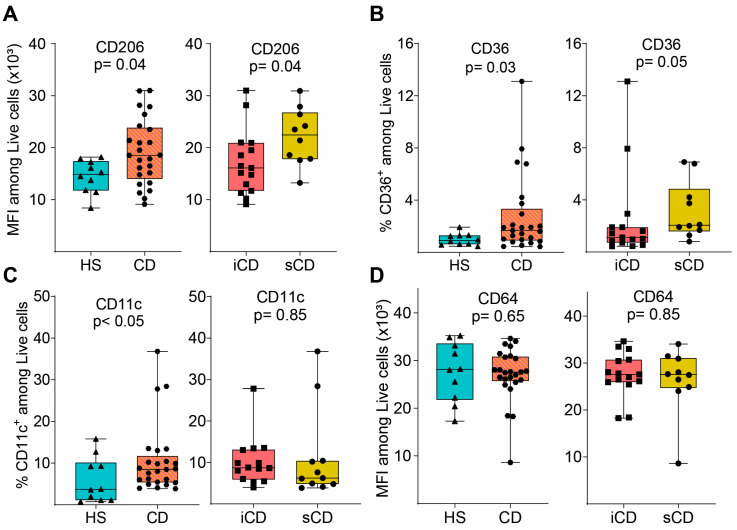
Surface marker expression of M2MQs treated with FS from patients with CD and HS. Macrophages were treated as in Figure 2A with FS from CD patients with inflammation (iCD, *n* = 15), CD patients with inflammation and stenosis (sCD, *n* = 10) and healthy subjects (HS, *n* = 10), and surface markers were analyzed by flow cytometry. Gating included live cells, single cells, and median fluorescent intensity (MFI) was determined for highly expressed markers while percentage was used for low expressed markers. Expression of (**A**) CD206, (**B**) CD36, (**C**) CD11c and (**D**) CD64 is shown. Significance was calculated by the Mann–Whitney U test. For each biological replicate, technical duplicates were pooled before analysis.

**Figure 4 cells-13-00060-f004:**
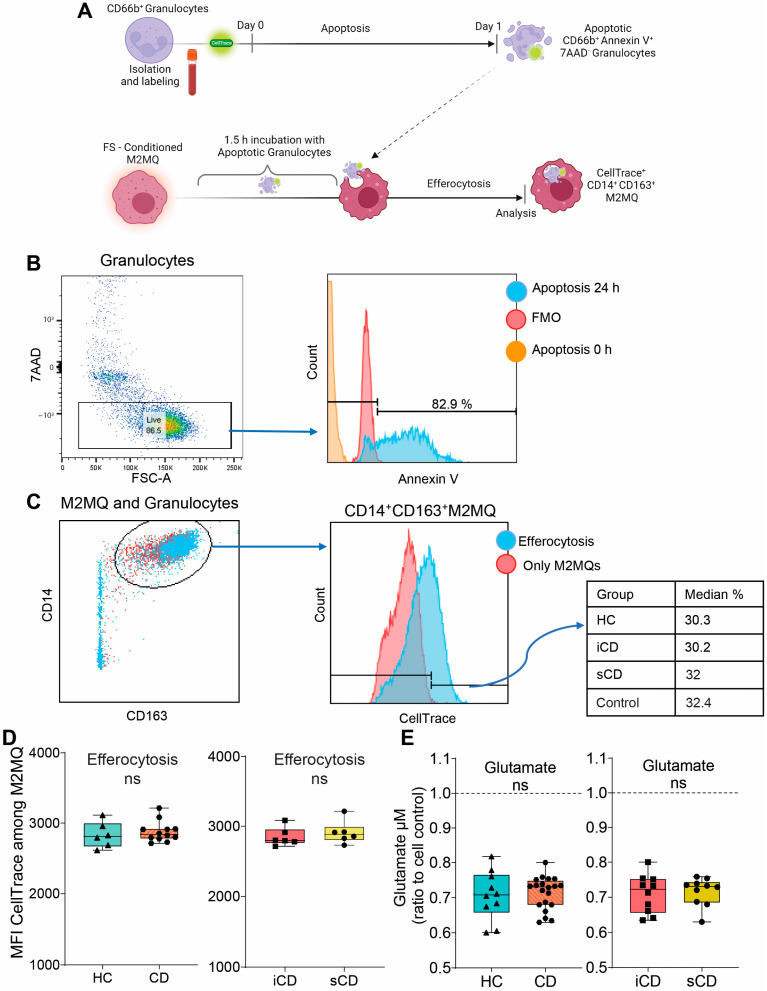
The efferocytotic function of M2MQs treated with FS from patients with CD and HS. (**A**) Schematic outline of the efferocytosis assay. Purified CD66b^+^ blood granulocytes were labeled with a fluorescent marker (CellTrace Violet) and spontaneous apoptosis was induced by 24 h incubation. Apoptotic granulocytes were cultured with M2MQs, obtained as shown in Figure 2A, for 1.5 h to induce efferocytosis. The macrophages were labeled with anti-CD14 and anti-CD163 and co-localization with CellTrace Violet was determined by flow cytometry. (**B**) Apoptotic CD66b^+^ granulocytes were assessed for phosphatidylserine translocation via AnnexinV binding. Gating of apoptotic cells included live 7AAD^−^ cells (left) and AnnexinV^+^ cells (right). FMO control was used to determine the gate. Repeated experiments from different granulocyte donors yielded reproducible results. (**C**) M2MQs were gated for CD14 and CD163 and the intensity of CellTrace Violet, showing engulfment of granulocytes, was analyzed. Plots show M2MQs incubated with (blue) or without (control; red) granulocytes. The average percentage of positive cells were comparable for all groups. (**D**) Median fluorescence intensity (MFI) of CellTrace Violet among CD14^+^CD163^+^ M2MQs comparing efferocytosis of macrophages conditioned with FS from patients with CD vs. healthy subjects (HS, *n* = 6) as well as CD patients with inflammation (iCD, *n* = 6) vs. CD patients with inflammation and stenosis (sCD, *n* = 6). The efferocytotic experiment was repeated for some of the FS samples, and it provided consistent results between samples. The experiment included randomly selected biological replicates, but no technical replicates, due to the limited number of granulocytes obtained. (**E**) Glutamate concentration as a measure of glutamine consumption, measured using luminescence, in cell culture supernatants of FS-conditioned macrophages from HS and CD (left) and iCD and sCD (right). FS from iCD (*n* = 10), sCD (*n* = 10) and HS (*n* = 10) were used. Non-significance is defined by ns.

**Figure 5 cells-13-00060-f005:**
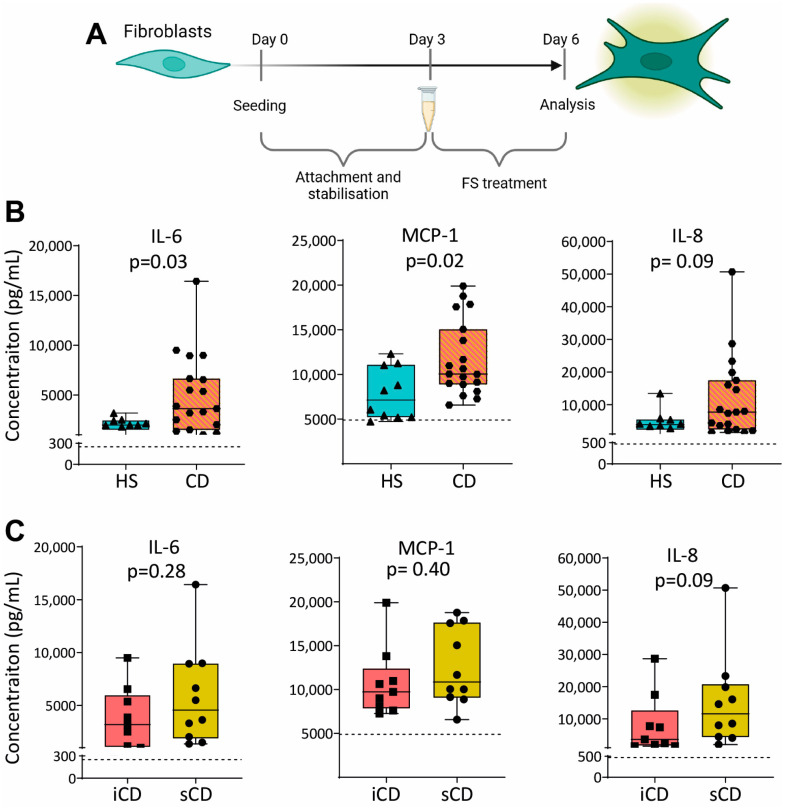
Cytokine secretion in fibroblasts treated with FS from patients with CD and HS. (**A**) Fibroblasts were cultured for 3 days without treatment to allow for the stabilization and attachment of the cells. At day 3, cells with FS, diluted 1:1000, was added to the cells. FS from CD patients with inflammation (iCD, *n* = 9), CD patients with inflammation and stenosis (sCD, *n* = 10) and healthy subjects (HS, *n* = 10) were used. On day 6, supernatants were analyzed by ELISA. Secretion of IL-6, MCP-1 and IL-8 from fibroblasts treated with FS from (**B**) HS vs. CD and (**C**) iCD vs. sCD is shown. For each biological replicate, technical duplicates were pooled before analysis. The dotted lines show protein secretion from untreated fibroblasts (control).

**Figure 6 cells-13-00060-f006:**
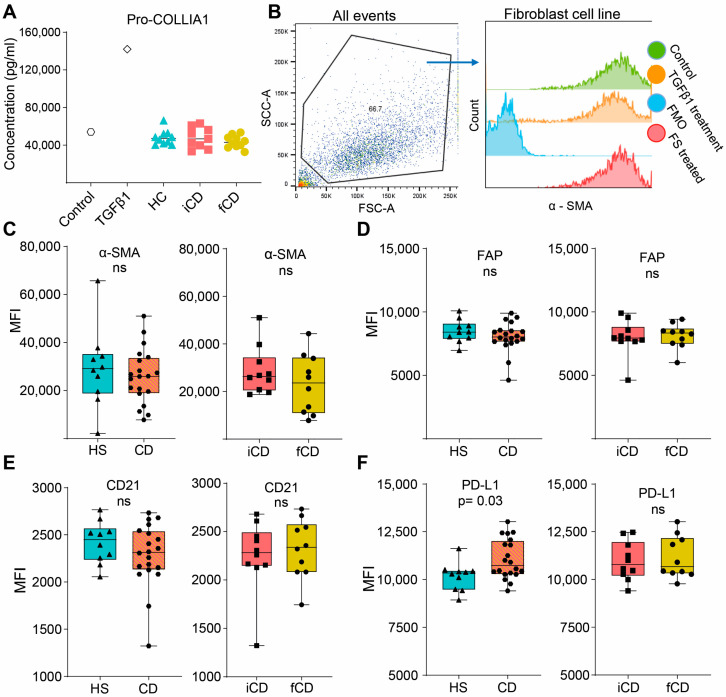
Fibrosis-related markers in fibroblasts treated with FS from patients with CD and HS. Fibroblasts were cultured and stimulated with FS, as shown schematically in Figure 5A, and supernatants and cells were analyzed using ELISA and flow cytometry, respectively. (**A**) Pro-COLLIA1 concentrations in supernatants are shown for Control (untreated cells), cells treated with 5 ng/mL TGFβ1 (positive control) and cells treated with FS. FS from CD patients with inflammation (iCD, *n* = 9), CD patients with inflammation and stenosis (sCD, *n* = 10) and healthy subjects (HS, *n* = 10) were used. (**B**) Gating strategy for α-SMA expression of control, TGFβ1-treated and an FS-treated sample (from a healthy subject). Fluorescence minus one (FMO) for α-SMA is shown for reference. Median fluorescence intensity (MFI) among α-SMA^high^ cells for α-SMA (**C**), FAP (**D**), CD21 (**E**) and PD-L1 (**F**) comparing FS from HS (*n* = 10) vs. patients with CD (*n* = 20) and patients with iCD (*n* = 10) vs. patients with sCD (*n* = 10) are shown. Non-significance is defined by ns.

**Figure 7 cells-13-00060-f007:**
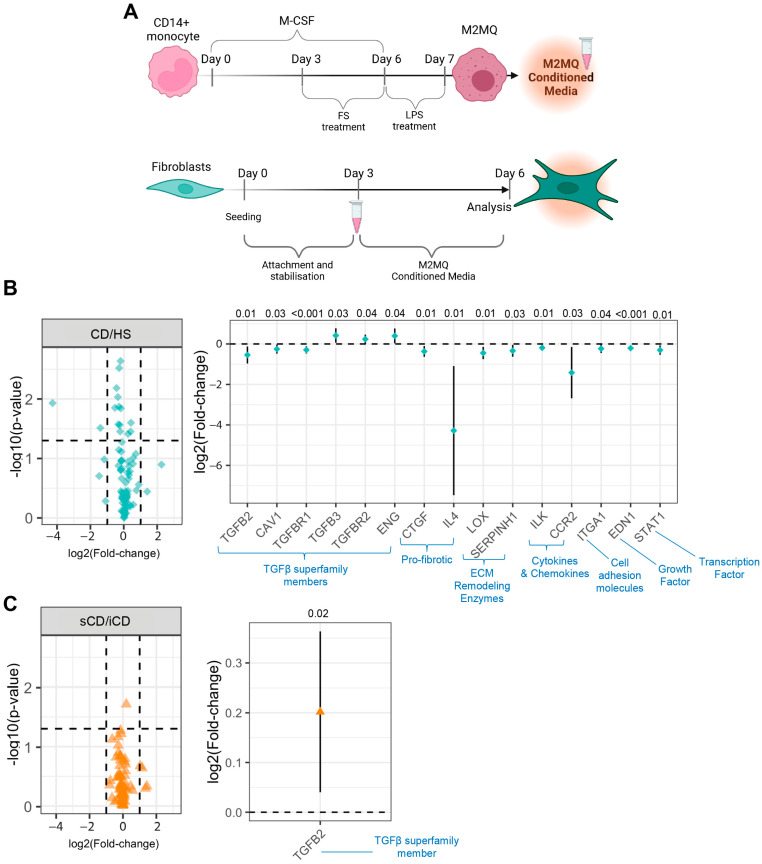
Fibroblasts cultured in the presence of conditioned media from M2MQs treated with FS from patients with CD and HS. (**A**) CD14+ blood monocytes were matured into CD14^+^CD163^+^CD80^−^ M2MQs with M-CSF for 6 days, with or without FS, diluted 1:1000 during days 3–6. FS from CD patients with inflammation (iCD, *n* = 7), CD patients with inflammation and stenosis (sCD, *n* = 7) and healthy subjects (HS, *n* = 6) were used. On day 6, cells were stimulated with 100 ng/mL LPS for 24 h and supernatants were collected (conditioned media). Fibroblasts were cultured for 3 days to allow for stabilization, and on day 3, media was changed to 50% fresh culture media and 50% M2MQ conditioned media. On day 6, cells were collected for gene expression analysis using a RT2 PCR array containing 84 fibrosis-related genes. Volcano (left) and forest (right) plots showing differentially expressed genes of fibroblasts cultured with conditioned media from M2MQs treated with FS from (**B**) CD compared to HS and (**C**) iCD compared to sCD, respectively. Conditioned media were produced in 5 replicates per biological replicate. The 5 replicates were pooled, and fibroblasts were in turn stimulated in 5 replicates which were pooled before RNA extraction. Forest plots significantly differently expressed genes (*p* < 0.05) with 95% confidence intervals. Significance was calculated by model-based *t* test. The dotted lines in (**B**,**C**) denote threshold for significance (horizontal) and two-fold up- or down-regulation (vertical).

**Table 1 cells-13-00060-t001:** Demographic and clinical parameters of patients with Crohn’s disease and healthy subjects.

	iCD (*n* = 15)	sCD (*n* = 10)	HS (*n* = 10)	*p*-Value ^3^
Sex (male/female)	7/8	4/6	6/4	0.74 ^4^
Age (years) ^1^	31 (20–59)	28 (20–53)	27 (21–64)	0.49 ^5^
Disease duration (years) ^1^	1 (0–8)	5 (0–24)	NA	0.04 ^5^
Disease behavior (colonic/ileocolonic/ileocecal)	5/9/1	5/2/3	NA	0.10 ^4^
Fecal calprotectin (mg/L) ^1^	700 (190–2600)	240 (110–6000)	<30 ^2^	0.24 ^5^
Blood CRP (mg/L) ^1^	5 (<1–36)	13 (<1–46)	1 (<1–7)	0.19 ^5^
HBI ^1^	7 (5–11)	7 (5–10)	NA	0.75 ^5^
CDEIS ^1^	14 (6–30)	15 (7–49)	NA	0.30 ^5^
Treatments Azathioprine (yes/no/previously) Infliximab (yes/no/previously)	2/12/1 2/13/0	6/2/2 3/7/0	NA	0.01 ^4^ 0.30 ^4^

^1^ Continuous data are shown as median (range). ^2^ One HS had calprotectin = 185 mg/kg. ^3^
*p*-values for comparison of iCD vs. sCD are shown. For comparison between CD and HS, the values were age *p* = 0.26, gender *p* = 0.39, calprotectin *p* < 0.0001 and CRP *p* = 0.03. ^4^ Chi-Square test. ^5^ Mann–Whitney U test. CD, Crohn’s Disease; iCD, patients with CD with inflammation; sCD, patients with CD with fibrosis and inflammation; HS, Healthy Subjects; NA, Not Applicable; CRP, C-Reactive Protein; HBI, Harvey-Bradshaw Index; CDEIS, Crohn’s Disease Endoscopic Index of Severity.

**Table 2 cells-13-00060-t002:** Fibrosis-related protein expression from fibroblasts cultured with conditioned media from FS-treated macrophages.

Protein	CD (*n* = 20)	HS (*n* = 10)	*p*-Value ^1^	iCD (*n* = 10)	sCD (*n* = 10)	*p*-Value ^1^
Pro-COLA1 (ng/mL) ^2^	92 (70–105)	86 (67–105)	0.15	97 (85–105)	87 (70–100)	0.03
Fibronectin (pg/mL) ^2^	323 (272–399)	322 (296–389)	0.70	324 (273–338)	322 (272–399)	0.90
FAP (MFI × 10^3^) ^3^	7.5 (6.7–8.5)	7.1 (6.2–8.2)	0.35	7.6 (6.7–8.5)	7.4 (6.7–7.7)	0.15
PD-L1 (MFI × 10^3^) ^3^	1.6 (1.4–2.1)	1.6 (1.4–1.9)	0.90	1.6 (1.4–2.1)	1.6 (1.4–1.9)	0.70
CD21 (MFI × 10^3^) ^3^	0.48 (0.43–0.59)	0.49 (0.47–0.53)	0.40	0.47 (0.43–0.52)	0.50 (0.46–0.59)	0.01
TGFb1 (MFI × 10^3^) ^3^	1.6 (1.1–2.0)	1.7 (1.2–1.8)	0.60	1.6 (1.1–2.0)	1.6 (1.5–2.0)	0.50

Continuous data are shown as median (range). ^1^ Mann–Whitney U test. ^2^ Determined by ELISA in the supernatants. ^3^ Determined by extracellular (FAP, PD-L1, CD21) and intracellular (TGFβ1) flow cytometry; gating is shown in Appendix A. CD, Crohn’s Disease; iCD, patients with CD with inflammation; sCD, patients with CD with stenosis and inflammation; HS, Healthy Subjects; MFI, median fluorescent intensity.

## Data Availability

The data presented in this study are available on request from the corresponding author.

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
