# Peer review of "Fecal Supernatants from Patients with Crohn’s Disease Induce Inflammatory Alterations in M2 Macrophages and Fibroblasts"

_cells, 2023, doi:10.3390/cells13010060_

Round 1
Reviewer 1 Report
Comments and Suggestions for Authors
Authors study the effect of fecal metabolites from Crohn´s disease (CD) patients on macrophages and fibroblasts by in vitro experiments. Macrophages derived from healthy donor peripheral blood monocytes and fibroblasts from a human colonic cell line are used for the study. They compared the composition of fecal metabolites from patients with inflammation, CD patients with inflammation and fibrosis, and healthy donors with results similar to previous studies. Additionally, they observed that fecal supernatants from CD patients promoted an anti-inflammatory protein profile of differentiated macrophages and a pro-inflammatory protein profile in fibroblasts.
Scientific background and rationale of the study as well as the research design. The presentation of the results is well described. The study is also properly referenced. This work completes previous studies of the group already published.
Although all the results are based on in vitro experiments with macrophages differentiated from monocytes and a colonic cell line, the study's approach is very interesting opening future research with colonic IBD samples. Additionally, future possible improvements of the therapies related to the diet and the microbiota.
Major points
Dot plots/histograms/overlays with the gating strategy and phenotype following monocyte and M2MQ differentiation should appear in Material and Methods (line143) or in Figure 3
Dot plots/histograms/overlays with the gating strategy and phenotype of the markers of fibroblast (FAP, PD-L1, CD21, and TGFB1) are necessary for Figure 7 or explaining Table 2
Minor points
They should comment on CD11c results in Figure 3.
Line 74; FS should be defined here
In line 36, a reference with a review of IBD is necessary
Reference 2; authors should choose a recent reference
Line 80; table 1 instead of Table 2
Line 250; FAP should be defined as “Fibroblast Activation Protein”
Line 254, alfa-sma should be defined as “α-smooth muscle actin”
Line 379; a space in necessary “forall”
Figure 7; “monocytes” term is necessary in the panel A
Line 465, “Authors” should not appear
Reviewer 2 Report
Comments and Suggestions for Authors
This manuscript shows that healthy subjects- or Crohn’s disease patients-derived fecal supernatants induce the pro-inflammatory cytokines in M2MQs, and thereby, treated M2MQs affect fibroblasts through their metabolites. This study is well-designed and assessed the effect of fecal ingredients on M2MQs and fibroblasts in vitro. I would like to suggest a couple of revisions below. Please check the comments and modify your manuscript.
Major comments:
1) L288 – 289: The authors divided CD patients FS into 2 groups, active inflammation (iCD) and stenosis (sCD), in this study. Could you show the rationale for why the authors divided into these two groups?
2) Several data in Fig. 2 and Fig. 3 are indicated by the ratio of FS treated over FS untreated samples (control), however, these are complicated for readers to interpret. Could you show the raw data (or figures plotted with raw data) in these relative data, such as Fig. 2C and Fig. 3A.
3) The authors show the efferocytosis were not affected by treatment with FS in Fig. 4. I am wondering how about the production of cytokines from M2MQs after the efferocytosis. Did the secretion of cytokines alter in this experiment?
Minor comments:
1) L364 – L365: Insert the reason why the authors measured glutamate levels in cell culture supernatants should be helpful to readers.
2) L394 – L395: Could you put the p-value in Fig. 5C (IL-6 and MCP-1)?
Round 2
Reviewer 2 Report
Comments and Suggestions for Authors
Thank you for replying to my comments. I agree with publishing this article.